# Measuring Corporate Digital Transformation: Methodology, Indicators and Applications

Limin Zou [1,2], Wan Li [1,*], Hongyi Wu [1,3], Jiawen Liu [4] and Peng Gao [1]

1  School of Economics and Management, University of Chinese Academy of Sciences, Beijing 100190, China; 13311362010@126.com (L.Z.); wuhongyi23@mails.ucas.ac.cn (H.W.); pgao@ucas.ac.cn (P.G.)
2  Kezhu Property Company, Administration Bureau of the Chinese Academy of Sciences, Beijing 100086, China
3  Sino-Danish College, University of Chinese Academy of Sciences, Beijing 101408, China
4  Economics and Management School, Wuhan University, Wuhan 430072, China; Chrissywen0116@163.com
*  Correspondence: liwan23@mails.ucas.ac.cn

**Abstract:** With the rapid development of data science, digital technology is integrating deeply with enterprise management, driving companies towards digital transformation to achieve sustainable development. However, digital transformation is a systematic and comprehensive process, posing challenges in accurately depicting firm-level digitalization. Hence, this study systematically reviews measurement methods for digital transformation across various themes related to enterprise digitalization. Existing literature predominantly employs questionnaire analysis, quantitative statistics, and text analysis to gauge the extent of digital transformation. In terms of indicator construction, existing literature mainly relies on input, process, and outcome variables to construct measurement indicators. Nevertheless, due to the subjectivity of questionnaires, the uniqueness of industry data, and the limitations of textual information, these methods and the indicators derived from them possess distinct applicability scopes. Refining the measurement of digital transformation should hinge on both the research objectives and the characteristics of the data. Furthermore, through the analysis of industry cases such as agriculture, manufacturing and service industries, this study also reveals the unique characteristics encountered by these industries in the process of digital transformation, provides a more detailed summary of measurement methods for these specific industries, and emphasizes the importance of selecting measurement methods according to industry characteristics.

**Keywords:** digital transformation; measurement methods; digital indicators; systematic review

## 1. Introduction

Over the past few decades, the theory and practice of corporate management have continuously evolved to adapt to ever-changing society, technology, and markets. In recent years, the rapid development of digital technology has brought profound changes to organizational management and business activities in the digital age. The widespread adoption of digital technologies like artificial intelligence, big data, cloud computing, and the internet of things has not only presented new opportunities for corporate development but also brought new challenges to enterprises such as data silos, resource coordination, and fulfillment of social responsibilities [1–4]. The management concepts and operational models that were originally established in the industrial era are now facing the impact of the digital age, compelling firms to embark on digital transformation. Digital transformation is not merely a means to keep pace with the times, but rather an inevitable option for firms to enhance their competitiveness and adapt to the rapidly changing market environment [5]. Enterprises must adopt new management concepts and business processes to adapt to the challenges brought by digital transformation [6]. Digital transformation for enterprises encompasses not only technological upgrades but also profound changes in organizational culture, leadership capabilities, strategic decision-making, and more [7–9]. This

holistic transformation not only affects the enterprises themselves but also has profound implications for the industry chain and even the entire socio-economic system [10,11].

Digital transformation has emerged as a crucial trend in modern business development. An increasing number of firms are leveraging digital technology to transform and optimize their business processes, promote operation efficiency, expand their market share, and generate greater customer value [12–16]. Existing literature primarily focuses on the influencing factors and the economy consequences of corporate digital transformation [17,18], providing comprehensive analyses about the various stages of digital transformation from the perspectives of internal organizational innovation and external industry revolution [19,20].

Digital transformation is closely intertwined with sustainable development, as enterprises utilize digitalization to adapt to ever-changing environments, expand their business horizons, and harness knowledge and technological resources to better address potential future risks and challenges, thereby achieving sustainable development [21]. Du and Qu [22] observed that the investment and application of digital technologies enable enterprises to foster the creation of sustainable development value. Additionally, research has indicated that corporate digital transformation can facilitate green technology innovation [23], enhance supply chain performance [24], generate synergistic effects in carbon reduction [25], and hold significant implications for the development of a low-carbon economy.

However, digital transformation is a systematic and comprehensive process, making it challenging to accurately describe digitalization at the firm level [26]. The absence of unified standards and norms, coupled with variations between industries and firms, further complicates the selection of measurement methodologies and indicators. Currently, there are significant differences in how corporate digital transformation is measured [27,28]. However, existing literature in this area is fragmented, and the measurement methods represented by dictionary method are also lacking in scientificity and accuracy [26]. A certain method may have inevitable disadvantages in other application scenarios. Different studies employ diverse measurement methodologies, lacking a systematic overview of the commonalities and characteristics of these measurement methods.

This paper aims to address the methodology, indicators, and applications involved in measuring the digital transformation of firms. By doing so, it will provide valuable references for researchers and practitioners, enabling them to gain a deeper understanding of and more effectively evaluate the digital transformation of firms.

In this review, we sourced literature mainly from the CSSCI for Chinese articles and the Web of Science for English articles, focusing on research from the past four years. We prioritized empirical studies and case analyses that measure corporate digital transformation outcomes. Exclusions were made for off-topic literature, studies with unclear methods, or those of lower quality, ensuring the reviews' relevance and scientific integrity. This approach helped systematically organize and analyze the latest developments in the field.

## 2. Measurement Methodology

### 2.1. Questionnaire Scale Method

Questionnaire surveys are commonly used to measure the level of digital transformation in firms by analyzing the responses obtained from scales or questionnaires. For instance, Dai et al. [29] assessed the data-based integration capability of firms with 172 questionnaires collected from managers. Similarly, Chi et al. [30] distributed questionnaires to 207 small and medium-sized medical device manufacturers in Hubei Province. They used evaluation indicators such as digital technology-based operations, integration, and transformation to measure the digital transformation of these firms and obtained first-hand survey data. In a study conducted by Yang et al. [31], a questionnaire consisting of 14 questions covering seven types of digital technologies was designed. They collected 615 questionnaires nationwide to measure the level of digital technology adoption and digital applications within firms. In another study by Tao et al. [32], the researchers enlisted the assistance of government officials in distributing questionnaires to company executives

and employees. This collaborative effort significantly enhanced the recovery rate and effectiveness of the questionnaires. In November 2020, a comprehensive survey on the digital development of over 22,000 private firms in 31 provinces, municipalities, and autonomous regions in China was conducted along with the 14th survey of private firms in China. This survey was jointly carried out by the United Front Work Department of the CPC Central Committee, the All-China Federation of Industry and Commerce, the State Administration for Market Regulation, the Chinese Academy of Social Sciences, and the China Institute of Private Sector. Many scholars have utilized the five stages of digital transformation as evaluation indicators in their questionnaires [33,34].

The questionnaire survey method is currently the main measurement method for the few foreign countries to quantitatively study the degree of digital transformation of enterprises. Crișan et al. [35] enhanced the questionnaire on digital transformation for a specific industry. They designed three questions for Romanian management consulting companies, focusing on the proportion of digital business, daily usage, and application of various technologies. Nadia et al. [36] distributed and collected 368 questionnaires to CEOs of small and medium-sized firms in the UAE and used scales consisting of five dimensions (i.e., information literacy, interaction, digital creation, and security) to evaluate the degree of digital transformation. Perera et al. [37] conducted a questionnaire survey on companies in the Australian design and construction industry based on their developed phased digital maturity framework. Belén et al. [38] proposed a DP2 indicator method that could comprehensively capture the level of digital maturity of a company. They invited randomly selected Spanish company managers to complete the questionnaire survey, focusing on eight dimensions: digital skills and technology application, digital management intensity, digital business processes, digital innovation performance, environmental performance, digital management and department agility, digital vision, and digital orientation. Other scholars, such as Chatterjee and Mariani [39] and Martínez [40], adapted the scale and placed greater emphasis on exploring how digital transformation could support or permeate the value chain to attract customers and facilitate business development.

In addition to questionnaires, semi-structured interviews are commonly used to gather information about the digital transformation of firms. A qualitative method with iterative coding process has emerged as a method to analyze unstructured data obtained from interviews or questionnaires in research on measuring digital transformation. This qualitative method allows for a broad understanding and defines the fundamental concept of digital transformation and enables researchers to deeply analyze the subjective perspectives of participants and the underlying beliefs [41]. The application of this coding process in the measurement of digital transformation follows the steps of grounded theorys' three-level coding approach, starting from open coding or pre-set coding of terms related to digital transformation in the interviews. Similar patterns are then clustered to generate axial coding, and finally, selective coding is employed to condense core categories [42–44].

The questionnaire method can study the stages and details of digitalization by designing targeted questions, but the response rate, completion rate and coverage of questionnaire survey can be relatively low, which may lead to insufficient samples. In addition, the topic setting of the questionnaire and the respondents' answers are influenced to some extent by subjective factors including cognition, emotion, and personal characteristics. Additionally, tracking the dynamic nature of digital transformation over time can be challenging through questionnaires alone.

### 2.2. Quantitative Statistical Method

The quantitative statistics method is another approach used to measure digitalization by analyzing relevant historical data that reflects the process of digital transformation in firms. This method involves examining both financial and non-financial data. Financial data are often sourced from the firms' annual reports. One widely used indicator to measure the degree of digital transformation is the investment in digital assets. Many studies have utilized the proportion of intangible assets related to the digital economy, as disclosed in the

detailed items of intangible assets in the notes to the financial reports of listed companies, to describe the level of digital transformation [45–47]. Specifically, when the detailed items of intangible assets in financial reports contain terms related to digital technology (e.g., "network", "client", "management system", "intelligent platform"), and related patents, these items are marked as "intangible assets of digital technology" and aggregate annual data [45]. In another study, Chen et al. [47] combined the text analysis method with deep learning models to refine the identification of digital intangible assets. They extracted basic phrases from policy documents, used deep learning models to filter and expand keywords, and finally retrieved financial reports based on the digital dictionary they created.

In addition to financial data, researchers have also utilized non-financial data to measure the level of digitization in firms. Bai and Yu [48] developed a scoring mechanism to evaluate the digitization level of companies based on terms related to digitization found in business registration information. This approach allows for a quantitative assessment of the extent of digitization in these firms. Furthermore, project investments that reflect digital transformation efforts, such as enterprise resource planning (ERP) and product lifecycle management (PLM) systems, have been used as indicators to measure the level of digitization [49]. Niu et al. [50] measured the level of digital outcomes in conglomerate enterprises, using the number of software copyrights related to digital technology owned by their digital technology subsidiaries.

To capture digitization in different industries, Liu et al. [51] took into account the content and process of digitization. They used the rates of investment in hardware and software information equipment and expenditures on electronic information and networks as indicators of digitization in the categories of information equipment and information networks, respectively. Furthermore, Liu et al. [51] selected alternative indicators to capture the digitization of operational process-centric firms. These indicators included the computer usage rate of employees, internet product sales rate, and ERP usage. By examining these variables, researchers can assess the extent to which operational processes have been digitized within firms. In the case of manufacturers, He and Wang [52] focused on digital transformation by selecting investment projects in digital hardware and software from the detailed items of fixed assets and intangible assets. They calculated the proportion of these investments and used the sum of the logarithmically transformed values of the two as a variable to represent digital transformation. Guo and Zhu [53] adopted a similar approach, measuring digital investment by the ratio of the net value of software and hardware investments in fixed assets to the total net asset value, further characterizing the degree of digital transformation.

From the above, the use of numerical data for quantitative analysis provides an objective and intuitive approach to measuring digital transformation. However, it is important to consider the specific characteristics of different industries, such as business models, market environments, and evaluation standards.

The universality of historical data may vary across different industries. Historical data collected from one industry may not directly apply to another industry due to these inherent differences. Therefore, when analyzing historical data for a specific industry, researchers may need to make specific adjustments or introduce industry-specific indicators to ensure the accuracy and relevance of the analysis.

### 2.3. Text Analysis Method

In recent years, with the extensive application of text big data in the social sciences, text analysis method has gradually become the mainstream method to describe the digital transformation of firms. However, there are variations in the specific processes of text analysis in different studies. For example, Wu et al. [54] differentiated between two levels: "underlying technology application" and "practical technology application" in their study. They constructed a feature word library for digital transformation based on classic academic literature and important policy documents. This feature word library was then utilized in the collection and organization of annual report texts from Chinese listed companies

using Python web scraping techniques. The measurement of digital transformation in firms was conducted by aggregating the word frequencies obtained through this process. In Zhao's study [55], the researcher employed a specific approach to construct digital transformation variables by analyzing annual reports. He first selected representative firms with notable digital development across different industries. Then, he segmented the annual reports of these firms to identify high-frequency words related to digital transformation. Additionally, he extracted text combinations of high-frequency words from the annual reports of listed firms in the overall sample as a supplementary method. Lastly, he used a word segmentation dictionary they created to count the frequency of keyword disclosures. These two methods for constructing digital transformation variables have gained recognition and have been referenced in subsequent studies [56–61]. Many scholars have made continuous advancements in text analysis methodologies, tailoring them to their specific research scopes and characteristics. For instance, they create feature word libraries for digital transformation by leveraging various materials such as corporate annual reports, industry research reports, and expert interviews. After segmenting the samples and calculating word frequencies, these scholars construct digital transformation indicators by employing the ratio of the total disclosure frequency to the total number of words in the corresponding annual report [58,62,63]. To address the "right-skewed" nature commonly observed in word frequency data, some researchers have applied logarithmic transformations to better characterize digital transformation. For example, they may algorithmize the frequency values by taking the natural logarithm of the sum of word frequencies plus one [59,61,64]. Wang et al. [65] differentiated and characterized the digital transformation strategies of retail companies in terms of depth and breadth. They measured the depth of transformation by the proportion of a companys' keywords to the total number of keywords of all companies in that year. Meanwhile, the breadth was represented by the number of different types of keywords related to digital transformation appearing in the annual reports.

Differences among various text analysis methods primarily arise from the selection of keywords and the scope of the textual data used for statistical analysis. While annual reports of listed firms are commonly used as a significant source for constructing word libraries, some scholars also extract information from national policy documents and news sources. For instance, Yuan et al. [63] manually segmented and identified 30 national policy documents related to the digital economy. They then identified 197 key words associated with corporate digitalization from these documents. This approach expands the sources of keywords beyond annual reports, providing a broader perspective on digital transformation. Furthermore, unlike other scholars who analyze the full text of annual reports, Yuan [63] and Xiao [66] focused specifically on the "Management Discussion and Analysis" (MD&A) section of the annual reports of listed companies. They argue that analyzing the MD&A section offers advantages in terms of representativeness, relevance, timeliness, fairness, and stringency. Chen et al. [67] focused on four prominent advanced technologies, namely artificial intelligence (AI), blockchain, cloud computing, and big data, for their analysis of digital transformation. They conducted keyword searches related to these technologies and used a dummy variable called "USER" to determine the level of digital transformation in firms. Cai et al. [68] expanded the scope of technologies by including the internet of things (IoT) as a keyword, particularly in the banking industry. They characterized the digitization level of banks by analyzing the frequency of relevant terms obtained from Google News searches. Li [56] recognized the importance of extracting keywords with common characteristics when combining different textual materials. Their analysis process involved extracting common words from existing word libraries and policy documents. They also captured four-word phrases from the firms' annual reports, manually screened out low-relevance phrases, and calculated word frequencies. In addition to traditional textual sources like annual reports, researchers have also looked into other sources such as business performance conferences and statements from social media to gather information about a firms' digital transformation process [69,70].

As the discussion on the digital transformation of firms continues to deepen, some research teams and institutions have started to build comprehensive databases and indices to provide a systematic measurement of digital transformation. One such example is the work of Zhang and Jing [71], who utilized the comprehensive index of digital transformation available in the CSMAR database. The CSMAR database contains the Digital Transformation Research Database of Listed Companies in China. This database is a collaborative effort between CSMAR and the Intelligent Business and Tech Firm Management research team at East China Normal University. The comprehensive index developed by using the CSMAR database offers a more objective and comprehensive measure of digital transformation compared to word frequencies obtained through text analysis. By relying on this index, researchers can examine digital transformation at meso and macro levels, incorporating a broader range of objective data. This approach provides a more accurate and comprehensive reflection of the digitalization process, helping to mitigate the issue of right-skewed data observed in word frequency-based analyses. Zhao et al. [72] adopted the Digital Transformation Index for Commercial Banks from Peking Universitys' Digital Finance Research Center to assess digitalization in banks. Based on textual content and financial data from the annual reports of 136 commercial banks between 2010 and 2018, this index examines three dimensions: banks' understanding of digital finance, organizational aspects, and digital financial products.

Text analysis plays a crucial role in analyzing unstructured data and providing valuable insights into the content collected in free-text format. It enables researchers to gain a better understanding of the overall digitalization of firms. However, the accuracy of text analysis methods is heavily dependent on the quality and scope of the library used. One key challenge in text analysis is the controversy surrounding the interpretation of word frequency. While word frequency can provide insights into executives' perception of digital transformation, it may not fully capture the actual state of digitalization within a firm [47].

## 3. Index Construction

### 3.1. Input Index

Liu et al. [51] took into account the essence of incorporated digitalization in different industries. They used the investment rate of hardware and software information equipment under new fixed investment and the expenditure rate on electronic information and networks as indicators of digitalization for information equipment and information network categories, respectively. They also included employee computer usage rate, internet product sales rate, and ERP usage as alternative indicators for digitalization in operational processes. Liu et al. [49] considered ERP, MES/DCS, PLM, and other digital transformation investment projects as key embedded methods for implementing digital management. They selected the corresponding investment amount as an independent variable to depict the digital transformation of firms. Li et al. [73] employed the ratio of software and hardware investment to net assets as a measure of the level of digital investment. Simultaneously, they assessed the level of digital technology and the degree of transformation of business models based on the frequency of appearance of keywords related to digital transformation in annual reports. This approach was utilized to calculate the extent of digital transformation in enterprises.

### 3.2. Outcome Indicators

A companys' intangible assets may contain a high proportion of digital economy-related assets, such as software, patents, technologies, brands, etc. Therefore, many studies have utilized the proportion of intangible assets related to the digital economy, as disclosed in the detailed items of intangible assets in the notes to the financial reports of listed companies, to describe the level of digital transformation [45–47,74]. Specifically, when the detailed items of intangible assets in financial reports contain terms related to digital technology (e.g., "network", "client", "management system", "intelligent platform"), and related patents, these items are marked as "intangible assets of digital technology" and

aggregated annually [45]. In another study, Chen et al. [47] combined the text analysis method with deep learning models to refine the identification of digital intangible assets. They extracted basic phrases from policy documents, used deep learning models to filter and expand keywords, and finally retrieved financial reports based on the digital dictionary they created. Guo and Zhu [53] considered various aspects in calculating digital investments. For hardware investment, they took into account fixed assets such as "office and electronic equipment, computer equipment, information technology equipment, and communication equipment". Software investment considered "office software, apps and support systems, information management and operating systems, and various information platforms" to be software assets. To measure digital transformation in manufacturers, He and Wang [52] selected the stock of investment projects in digital hardware and software from detailed items of fixed assets and intangible assets. They calculated the proportion of these investments and used the sum of the logarithmically transformed values of the two as the variable for digital transformation.

### 3.3. Process Index

Digital transformation is a systematic strategic change, which not only involves the use of a variety of digital technologies, but also involves the deep integration of digital technologies with enterprise organization and management, and products/services. Recognizing the multidimensional nature of digital transformation, research teams and institutions have been developing comprehensive indicators and frameworks to characterize and assess the overall digital panorama of firms. These indicators aim to provide a more holistic view of digital transformation by considering multiple dimensions and aspects of the process. Wang et al. [75] developed a maturity evaluation model for digitalization in manufacturing firms based on the Analytic Hierarchy Process (AHP). The model consists of four dimensions: strategy, operational technology, cultural organizational capabilities, and ecosystem. Digital transformation is decomposed into 13 categories and 35 refined factor domains. The Digital Transformation Research Database of Listed Companies in China, a collaboration between CSMAR Database and the research team of "Intelligent Business and Tech Firm Management" at East China Normal University, has developed an enterprise digital index that provides a comprehensive assessment of digital transformation. This index incorporates six perspectives: strategic guidance, technology drive, organizational empowerment, environmental support, digital outcomes, and digital applications. In addition to measuring word frequency related to big data technology, the database includes objective indicators such as digital capital investment and invention patent applications in various industries. These objective indicators provide a more accurate and comprehensive reflection of the level of corporate digitalization, going beyond textual analysis and incorporating tangible measures of investment and innovation. Using text analysis and quantitative statistics, Dai and Ma [76] approached digital transformation from three aspects: macro-level digital industrialization, industry digitalization, and micro-level enterprise digitalization. They employed principal component analysis to reduce the dimensionality of secondary sub-indicators, ultimately deriving comprehensive measurement indicators for digital transformation. Niu et al. [50] quantified the overall level of digital transformation of A-share listed companies from three dimensions: digital investment, digital focus, and digital outcomes.

Compared to other industries, the measurement of digital transformation in the banking and manufacturing sectors has garnered more attention. In the banking sector, the Digital Finance Research Center at Peking University has developed the Digital Transformation Index for Commercial Banks. This index is constructed based on the analysis of annual report texts and financial data of 136 commercial banks spanning from 2010 to 2018. It encompasses three dimensions that capture different aspects of digital transformation in banks: banks' understanding of digital finance, organizational aspects, and digital financial products. Similarly, Xie and Wang [77] constructed a set of indicators system for the digital transformation of banks from three dimensions: strategic digitalization, business digitaliza-

tion, and management digitalization. These indicators include sub-indicators such as the frequency of digital technology mentions, channels, products, research and development, architecture, talent, and cooperation. They also flexibly utilize text analysis methods to comprehensively analyze publicly available information from banks.

It provides a detailed discussion on assessment such as strategy and organization, digital foundation, digital technology applications, business integration, enterprise comprehensive integration, and industry collaborative innovation. Wan et al. [78] have developed an assessment framework for digital transformation in the manufacturing industry, focusing on the dimensions of value, factor, and capability. The framework comprises three parts: strategy and foundation assessment, level and capability assessment, and effectiveness and benefit assessment. Chen and Xu [79] have constructed an evaluation system for the digital transformation capabilities of firms. The system consists of seven aspects and 26 indicators, including digital infrastructure, digital research and development, digital investment, organizational structure, digital talent, business digital management, production digital management, and financial digital management.

Although the comprehensive indicator system is more objective and comprehensive, it has certain limitations. First, on the practical level, the collection and collation of a large number of data has increased the research intensity; secondly, due to the differences in production and operation processes of different industries, it is difficult to apply the same set of indicators uniformly to measure digital transformation across a large number of firms simultaneously.

## 4. Industry Application Examples

In practical research, there are no absolute standards or norms for measuring corporate digital transformation. Given the diverse nature of industries and firms, measuring digital transformation requires flexibility and the selection of appropriate research methods based on data availability and indicator feasibility. This approach allows for an objective description of existing reality to the greatest extent possible. Additionally, due to the differences in business models, targets, and focus of digital transformation across industries, it is essential to consider the unique characteristics of different types of firms. Therefore, researchers often conduct studies that specifically focus on the digital transformation processes of firms in various industries or with different ownership types [80]. These studies may employ case study methodologies, presenting the digital transformation processes of specific firms through single or multiple cases. By examining these cases, researchers can propose targeted and specialized methods for measuring digital transformation that are tailored to the specific context and objectives of the firms being studied.

### 4.1. Primary Industry

Agriculture, as a fundamental industry crucial to national welfare and livelihood in the primary sector, plays a vital role in alleviating information asymmetry, enhancing the efficiency of value chains, and promoting sustainable environmental development [81].

Marc et al. [82] conducted a decade-long longitudinal multi-case study involving 65 semi-structured interviews with management personnel from two agricultural equipment manufacturers, along with their associated suppliers and data companies. The study also incorporated a range of internal and external data sources, including news reports, annual reports, application documents, and strategic planning files, initially applying open coding techniques aimed at uncovering the characteristics of various events and decisions. Additionally, the analysis of pricing strategies, developer information, download counts, and target customer data further enhanced the assessment of the stages of digital transformation within these enterprises. Yang and Cui [83] provided a detailed analysis of the digital transformation process of New Jinnong, a company based in Shenzhen, China, categorizing it into two phases, 1.0 and 2.0. They posited that the company would transition from digital 1.0 to 2.0 by establishing three major systems, metrics, and platforms. By evaluating the financial performance of the company across production, operations, supply

chain management, and value creation, it was determined that the company was in the initial stages of its digital transformation. Ciruela et al. [84] designed a novel diagnostic tool that enables agricultural cooperatives to conduct self-assessments of their digital transformation. This tool, based on questionnaire data from the European Statistical Office, covers multiple dimensions including human resources and management, internet usage, website operations, e-commerce, cloud computing, big data, the internet of things, automation, blockchain, and artificial intelligence. This diagnostic tool has been successfully applied in two agricultural cooperatives, effectively assessing their level of digital transformation.

### 4.2. Secondary Industry

The industrial sector serves as a driver of economic development, and its digital transformation is essential for enhancing production efficiency, reducing operational costs, and accelerating product innovation. Research primarily focuses on the manufacturing industry, with fewer studies investigating its sub-sectors.

Wen et al. [85] utilized input–output tables and information from corporate annual reports to construct an interaction term that multiplies the total consumption coefficient of digital products by the frequency of digital transformation-related terms in corporate annual reports. This interaction term serves as a proxy variable to gauge the extent of digital transformation in manufacturing enterprises. Wan et al. [86] developed a lexicon of 89 key terms based on China's top-level design documents for the digitalization of the manufacturing sector. This lexicon is used to extract the frequency of key terms from corporate annual reports. Yi et al. [87] created a questionnaire that covers four aspects of digitalization: production, management, logistics, and service, comprising seven items in total. Wu et al. [88] categorized the stages of enterprise digital transformation into initial, growth, and maturity phases. They derived key terms for each stage from an extensive review of the literature and public information, and employed text analysis to statistically analyze, summarize, and iterate the frequency of terms in corporate annual reports, thus aiding in the assessment of the enterprises' transformation stages.

### 4.3. Tertiary Industry

The tertiary sector, serving as a crucial engine for economic growth, has seen significant benefits from digital transformation, which helps businesses enhance service quality, increase customer satisfaction, and expand market share. The scope of the tertiary sector is quite broad, with related research on digital transformation in enterprises primarily focused on the banking, retail, and service industries.

#### 4.3.1. Banking

Digital technology has played a pivotal role in facilitating profound reforms and upgrades in the traditional business models of commercial banks. This digital transformation has had a positive impact on operational efficiency, service quality, and customer experiences. Various digital advancements, such as digital payment solutions, accurate credit identification systems, and intelligent investment and wealth management technologies, have significantly improved the efficiency of business processes and financial services within commercial banks. Furthermore, the banking industry has demonstrated a significantly higher degree of application of financial technology and progress in driving digital transformation compared to other financial sectors [89]. Many studies on digital transformation in the banking industry have employed various measurement methods.

Xiang and Gao [90] summarized the measurement methods of digital transformation in the banking sector and classified the measurement methods into five categories: annual report mining, news statistics, strategic cooperation, patent application, and digital product usage. Some other studies developed indices specifically for measuring digital transformation in the banking industry. Xie and Wang [77] developed the Digital Transformation Index for Chinese Commercial Banks at Peking University, which encompasses three dimensions: strategic digitalization, business digitalization, and management digitalization.

This index covers 228 banks with a total asset value of 206.49 trillion yuan at the end of 2018, accounting for 98.35% of the total assets of commercial banks in China, demonstrating strong industry representativeness.

### 4.3.2. Retail

The retail industry has traditionally been characterized by a strong physical presence. However, in recent years, brick-and-mortar retailers have been increasingly integrated with the virtual economy. This evolution has been driven by the constant transformation and reshaping of the traditional retail industry through digital technology. As a result, digital transformation in the retail industry has become an important area of research. Hagberg et al. [91] proposed a "transaction–product–participant–context" quadripartite conceptual framework to specifically analyze the digital transformation phenomenon in the retail industry. Wang et al. [92] conducted a multiple-case study on the digital transformation of retailers, covering both traditional retailers such as RT-Mart and Red Dragonfly, as well as emerging retailers such as Freshippo and Mi Home. The researchers collected data through semi-structured interviews, informal interviews, and secondary data collection. Subsequently, the data were cleaned, refined, and encoded to construct three indicators: digital infrastructure capability, digital governance capability, and the capability of overcoming digital divide and traps. These indicators were used to measure the level of digital transformation. Chaparro et al. [93] constructed a structured set of indicators from the consumer perspective to measure digital transformation in Spanish electricity retailers. These indicators include remote consumption monitoring, consumption and plan assessment, digital billing, digital customer data management, and customer touchpoints (channel offering). Chen and Cheng [94] improved the balanced scorecard performance evaluation system model, incorporating the characteristics of new retail. The value creation and delivery dimension examined multiple indicators at the information technology level of retail enterprises, including online addition conversion rate, online transaction conversion rate, research and development investment rate, app download rate, and app satisfaction.

### 4.3.3. Service Sector

Digital technology empowers firms to offer more precise and sophisticated services to users, allowing service providers to better understand customer needs and generate greater value. Gu and Zhang [95] conducted a study on the digital transformation process of Xibei Catering Group. They analyzed the digital maturity of the firm by examining various aspects, including infrastructure, human resources, purchasing and processing, marketing and services (internally), as well as consumers and partners (externally). Zhang et al. [96] provided a global view of the digital transformation process from a demand-driven human resources service model to an intelligent matching human resources service model using a specific human resources firm as an example. To gain insights into this transformation, the researchers collected data through in-depth interviews and focus group interviews. They conducted a detailed qualitative analysis of the HR firms' business resources, human resources, and data resources capabilities.

In summary, digital transformation varies across different industries, with each industry having its unique focus areas. Furthermore, the outcomes of digital transformation can vary among firms within the same industry. Consequently, it is crucial for researchers to take into account the characteristics of different types of firms and gather more specific and granular data. In recent years, research studies have increasingly adopted a detailed approach by conducting multiple or even single case studies to present in-depth digital transformation processes. Moreover, they have developed measurement methodology tailored to specific industries or even individual firms to capture the nuances of digital transformation more effectively.

## 5. Conclusions and Outlook

In the era of the digital economy, digital transformation has emerged as a crucial strategy for enhancing competitiveness and achieving sustainable development. This paper offers a comprehensive examination of methods, index construction, and their application in measuring the digital transformation of firms across different industries. The findings presented in this paper serve as a valuable reference for both the academic and the business communities. Through in-depth analysis of questionnaires, quantitative statistics, and text analysis, this study sheds light on the strengths and limitations of various measurement methods. It emphasizes the importance of selecting appropriate methods based on research objectives and data characteristics when applying them. By providing insights into the advantages and challenges associated with different approaches, this paper contributes to the development of more robust and effective methodologies for measuring digital transformation. Moreover, this article delves into the construction of input indicators, outcome indicators, and process indicators, offering a thorough assessment of digitalization. These indicators not only contribute to a comprehensive understanding of digital transformation but also provide fresh perspectives on comprehending its complexity and dynamics. Through the analysis of industry cases, such as the agriculture, manufacturing, and service industries, this article uncovers the distinct characteristics and challenges encountered in their digital transformation journeys. It emphasizes the significance of industry-specific characteristics when selecting measurement methods and indicators for assessing digital transformation. These case studies not only enrich the theoretical framework of digital transformation but also offer practical guidance for firms navigating their own digitalization efforts.

This paper also highlights the limitations of current research in measurement methods and index construction. These include issues such as data source discrepancies, challenges in tracking dynamic changes, and insufficient consideration of industry-specific factors. Given the rapid advancements in digital technology, there is a growing need for new measurement methods and indicators that can adapt to the ever-evolving technological and business landscape. Future research should prioritize the development of more accurate and universally applicable measurement tools to address these shortcomings.

Future research should be dedicated to developing more accurate and universally applicable measurement tools that are better suited to the constantly evolving technological and business environments while also supporting companies in achieving sustainable development through their digital transformation processes. As global considerations for environmental protection, social responsibility, and economic efficiency become increasingly important, new measurement tools need to be capable of assessing how digital transformation can enhance resource efficiency, reduce environmental impacts, and improve social welfare. For instance, future studies could explore how to leverage big data and artificial intelligence technologies to precisely track and evaluate corporate performance in areas such as carbon reduction, energy efficiency, and sustainable supply chain management. Additionally, research should focus on developing a cross-industry evaluation framework that includes a comprehensive assessment of corporate performance in environmental, social, and governance (ESG) aspects, aiding companies not only in achieving economic success but also in making positive contributions on social and environmental levels. Through these precise and universally applicable measurement tools, companies can better understand and manage the challenges and opportunities presented during their digital transformation, thereby advancing global progress toward sustainable development.

**Author Contributions:** L.Z., W.L. and H.W.: data preparation and design, original draft preparation, J.L.: formal analysis, manuscript preparation. P.G.: conceptualization and supervised this project. All authors have read and agreed to the published version of the manuscript.

**Funding:** This research is funded by National Natural Science Foundation of China (72334006).

**Institutional Review Board Statement:** Not applicable.

**Informed Consent Statement:** Not applicable.

**Data Availability Statement:** Data will be provided upon request.

**Conflicts of Interest:** The authors declare no conflicts of interest.

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
