# Peer review of "Measuring Corporate Digital Transformation: Methodology, Indicators and Applications"

_sustainability, doi:10.3390/su16104087_

Round 1
Reviewer 1 Report
Comments and Suggestions for Authors
In my opinion, the article is professional rather than scientific. The authors work with secondary research and present their point of view to the scientific public. I am missing clear methods, procedures for working with secondary data that would support chapter 5. It is not clear to me why the authors in chapter 4 focused only on three industry applications and why exactly the banking, retailing and service sector. In my opinion, there are other (more important) industrial sectors in which digitization is taking place. Chapter 5 is very blunt and I would expect a clear position of the authors on the topic in this chapter. I recommend reworking the article.
Reviewer 2 Report
Comments and Suggestions for Authors
Review of the paper Titled “Measuring corporate digital transformation: methodology, indicators and applications”
While the authors have written an interesting and timely review article on the measurement of corporate digital transformation, addressing the following comments for each section could further strengthen their contribution and manuscript
Abstract
Although the authors have written a good abstract capturing background, methods and aims, the authors could make it intact by adding a sentence or two about the specific outcomes of the research. Briefly mention the key findings or insights gained from the analysis.
Introduction
The current introduction, while effectively setting the context (lines 38-76), lacks citations to existing research. In academic writing, a strong introduction positions the current study within the existing body of scholarship. This is achieved by referencing relevant past studies and highlighting the gap your research aims to address. By incorporating citations and explaining the research gap, the introduction will effectively demonstrate the significance and novelty of your current study.
The authors need to re write the introduction with the following elements (background, position the study in existing studies, show the gaps that warrants your study to be carried out, argue for the contribution of the study, briefly touch on the need for a review on the topic (why should a review be carried out on the topic). Briefly highlight key and crucial findings unique to the study)
Methodology
The authors mentioned that they carry out a review. Please provide a paragraph justifying the need for a review. Secondly the authors did not explain how they included the literature used in their review, what process was used to select the studies, from where did you get the studies, what criteria determined the studies used in your review?
Given the importance of the topic, a mere review robs the paper of its quality and contributions.
The authors could make a very strong contribution by carrying out a systematic review methodology with a proper methodology section. This involves defining clear inclusion and exclusion criteria for relevant studies. Studies could be sourced from well-trusted databases (Scopus and Web of Sciences) as well as grey literature from Google Scholar. Selection criteria should specify publication date (e.g., past five years), research methodology (empirical studies preferred), and focus on measuring corporate digital transformation outcomes. Through this rigorous process, the review can provide a valuable roadmap for businesses navigating the complexities of digital transformation measurement.
Comments on the Quality of English LanguageGood English quality
Round 2
Reviewer 1 Report
Comments and Suggestions for Authors From my point of view, the authors incorporated the required things into the article.
Reviewer 2 Report
Comments and Suggestions for Authors
My only main concern which is not adequately addressed is that the introduction still lacks sufficient citations. Large sections of text were not supported by references.
From line 39 to line 57 there is a large text with many arguments but no proper credit and citations. This is an anomaly in academic writing. Please add proper citations where possible most recent.
Line 51 to line 64 is a huge section of text without proper citations. Please at least three citations
Line 74 to line 86 is also a huge text section without proper citations. Please add at least 3 citations.
Comments on the Quality of English Languageminor language editing
